# Data Diversification: A Simple Strategy For Neural Machine Translation

**Xuan-Phi Nguyen**[1,3]**, Shafiq Joty**[1,2]**, Wu Kui**[3]**, Ai Ti Aw**[3]
[1]Nanyang Technological University
[2]Salesforce Research
[3]Institute for Infocomm Research (I[2]R), A*STAR
Singapore
{nguyenxu002@e.ntu,srjoty@ntu}.edu.sg
{wuk,aaiti}@i2r.a-star.edu.sg

## Abstract

We introduce Data Diversification: a simple but effective strategy to boost neural machine translation (NMT) performance. It diversifies the training data by using the predictions of multiple forward and backward models and then merging them with the original dataset on which the final NMT model is trained. Our method is applicable to all NMT models. It does not require extra monolingual data like back-translation, nor does it add more computations and parameters like ensembles of models. Our method achieves state-of-the-art BLEU scores of 30.7 and 43.7 in the WMT'14 English-German and English-French translation tasks, respectively. It also substantially improves on 8 other translation tasks: 4 IWSLT tasks (English-German and English-French) and 4 low-resource translation tasks (English-Nepali and English-Sinhala). We demonstrate that our method is more effective than knowledge distillation and dual learning, it exhibits strong correlation with ensembles of models, and it trades perplexity off for better BLEU score.

## 1 Introduction

The invention of novel architectures for neural machine translation (NMT) has been fundamental to the progress of the field. From the traditional recurrent approaches [22, 14], NMT has advanced to self-attention method [24], which is more efficient and powerful and has set the standard for many other NLP tasks [3]. Another parallel line of research is to devise effective methods to improve NMT without intensive modification to model architecture, which we shall refer to as *non-intrusive extensions*. Examples of these include the use of sub-word units to solve the out-of-vocabulary (OOV) problem [18] or exploiting extra monolingual data to perform semi-supervised learning using *back-translation* [17, 4]. One major advantage of these methods is the applicability to most existing NMT models as well as potentially future architectural advancements with little change. Thus, non-intrusive extensions are used in practice to avoid the overhead cost of developing new architectures and enhance the capability of existing state-of-the-art models.

In this paper, we propose *Data Diversification*[1], a simple but effective way to improve machine translation consistently and significantly. In this method, we first train multiple models on both backward (target→source) and forward (source→target) translation tasks. Then, we use these models to generate a diverse set of synthetic training data from both lingual sides to augment the original data. Our approach is inspired from and a combination of multiple well-known strategies: back-translation, ensemble of models, data augmentation and knowledge distillation for NMT.

Our method establishes the state of the art (SOTA) in the WMT'14 English-German and English-French translation tasks with 30.7 and 43.7 BLEU scores, respectively.[2] Furthermore, it gives 1.0-2.0 BLEU gains in 4 IWSLT tasks (English↔German and English↔French) and 4 low-resource tasks (English↔Sinhala and English↔Nepali). We demonstrate that data diversification outperforms other related methods – knowledge distillation [13] and dual learning [27], and is complementary to back-translation [17] in semi-supervised setup. Our analysis further reveals that the method is correlated with ensembles of models and it sacrifices perplexity for better BLEU.

## 2   Background

**Novel Architectures**   The invention of novel neural architectures has been fundamental to scientific progress in NMT. Often they go through further refinements and modifications. For instance, Shaw et al. [19] and Ahmed et al. [1] propose minor modifications to improve the original Transformer [24] with slight performance gains. Ott et al. [15] propose scaling the training process to 128 GPUs to achieve more significant improvements. Wu et al. [28] repeat the cycle with dynamic convolution. Side by side, researchers also look for other complementary strategies to improve the performance of NMT systems, which are orthogonal to the advancements in model architectures.

**Semi-supervised   NMT**   Semi-supervised learning offers considerable capabilities to NMT models. Back-translation [17] is a simple but effective way to exploit extra monolingual data. Another effective strategy is to use pretrained models. Zhu et al. [30] recently propose a novel way to incorporate pretrained BERT [3] to improve NMT. Nonetheless, the drawback of both approaches is that they require huge extra monolingual data to train/pretrain. Acquiring enormous datasets is sometimes expensive, especially for low-resource scenarios (languages or domains). Moreover, in the case of using pretrained BERT, the packaged translation model incurs the additional computational cost of the pretrained model.

Table 1: Estimated method comparison. $|\Theta|$ denotes the number of parameters, while $|D|$ denotes the size of *actual* training data required.

| Method | Training | | | Inference | |
|---|---|---|---|---|---|
| | **FLOPs** | $|\Theta|$ | $|D|$ | **FLOPs** | $|\Theta|$ |
| *New Architectures* | | | | | |
| Transformer | 1× | 1× | 1× | 1× | 1× |
| Dynamic Conv | 1× | 1× | 1× | 1× | 1× |
| *Semi-supervised* | | | | | |
| NMT+BERT | > **60**× | 3× | > **25**× | **3**× | **3**× |
| Back-translation | 2× | 1× | > **50**× | 1× | 1× |
| *Evolution-based* | | | | | |
| So et al. [21] | > **15000**× | 1× | 1× | 1× | 1× |
| *Our Data Diversification* | | | | | |
| Default Setup | 7× | 1× | 1× | 1× | 1× |

**Resource Trade-offs**   Table 1 summarizes different types of costs for training and inference of different approaches to improve NMT. Developing new architectures, like dynamic convolution [28], offers virtually no measurable compromise for training and inference, but it may take time for new models to be refined and mature. On the other hand, semi-supervised methods are often simpler, but require significantly more training data. In particular, Edunov et al. [4] use back-translation with 50× more training data. NMT+BERT [30] requires 60× more computations and 25× more data to train (including the pre-training stage). It also needs 3× more computations and parameters during inference. Evolved-Transformer [21], an evolution-based technique, requires more than 15,000 times more FLOPs to train. This may not be practical for common practitioners.

On the other hand, our data diversification method is simple as back-translation, but it requires no extra monolingual data. It also has the same inference efficiency as the "*New Architectures*" approach. However, it has to make compromise with extra computations in training.[3]

## 3   Method

### 3.1   Data diversification

Let $\mathcal{D} = (S, T)$ be the parallel training data, where $S$ denotes the source-side corpus and $T$ denotes the target-side corpus. Also, let $M_{S \to T}$ and $M_{T \to S}$ be the forward and backward NMT models,

**Algorithm 1** Data Diversification: Given a dataset $\mathcal{D} = (S, T)$, a diversification factor $k$, the number of rounds $N$; return a trained source-target translation model $\hat{M}_{S \to T}$.

---

1: **procedure** TRAIN($\mathcal{D} = (S, T)$)
2:      Train randomly initialized $M$ on $\mathcal{D} = (S, T)$ until convergence
3:      **return** $M$
1: **procedure** DATADIVERSE($\mathcal{D} = (S, T), k, N$)
2:      $\mathcal{D}_0 \leftarrow \mathcal{D}$                                    ▷ Assign original dataset to round-0 dataset.
3:      **for** $r \in 1, \dots, N$ **do**
4:          $\mathcal{D}_r = (S_r, T_r) \leftarrow \mathcal{D}_{r-1}$
5:          **for** $i \in 1, \dots, k$ **do**
6:              $M_{S \to T, r}^i \leftarrow$ TRAIN($\mathcal{D}_{r-1} = (S_{r-1}, T_{r-1})$)          ▷ Train forward model
7:              $M_{T \to S, r}^i \leftarrow$ TRAIN($\mathcal{D}'_{r-1} = (T_{r-1}, S_{r-1})$)         ▷ Train backward model
8:              $\mathcal{D}_r \leftarrow \mathcal{D}_r \cup (S, M_{S \to T, r}^i(S))$              ▷ Add forward data
9:              $\mathcal{D}_r \leftarrow \mathcal{D}_r \cup (M_{T \to S, r}^i(T), T)$             ▷ Add backward data
10:     $\hat{M}_{S \to T} \leftarrow Train(\mathcal{D}_N)$                         ▷ Train the final model
11:     **return** $\hat{M}_{S \to T}$

---

which translate from source to target and from target to source, respectively. In our case, we use the Transformer [24] as the base architecture. In addition, given a corpus $X_l$ in language $l$ and an NMT model $M_{l \to \hat{l}}$ which translates from language $l$ to language $\hat{l}$, we denote the corpus $M_{l \to \hat{l}}(X_l)$ as the translation of corpus $X_l$ produced by the model $M_{l \to \hat{l}}$. The translation may be conducted following the standard procedures such as maximum likelihood and beam search inference.

Our data diversification strategy trains the models in $N$ rounds. In the first round, we train $k$ forward models $(M_{S \to T, 1}^1, \dots, M_{S \to T, 1}^k)$ and $k$ backward models $(M_{T \to S, 1}^1, \dots, M_{T \to S, 1}^k)$, where $k$ denotes a diversification factor. Then, we use the forward models to translate the source-side corpus $S$ of the original data to generate synthetic training data. In other words, we obtain multiple synthetic target-side corpora as $(M_{S \to T, 1}^1(S), \dots, M_{S \to T, 1}^k(S))$. Likewise, the backward models are used to translate the target-side original corpus $T$ to synthetic source-side corpora as $(M_{T \to S, 1}^1(T), \dots, M_{T \to S, 1}^k(T))$. After that, we augment the original data with the newly generated synthetic data, which is summed up to the new round-1 data $\mathcal{D}_1$ as follows:

$$\mathcal{D}_1 = (S, T) \bigcup \cup_{i=1}^k (S, M_{S \to T, 1}^i(S)) \bigcup \cup_{i=1}^k (M_{T \to S, 1}^i(T), T) \tag{1}$$

After that, if the number of rounds $N > 1$, we continue training round-2 models $(M_{S \to T, 2}^1, \dots, M_{S \to T, 2}^k)$ and $(M_{T \to S, 2}^1, \dots, M_{T \to S, 2}^k)$ on the augmented data $\mathcal{D}_1$. The similar process continues until the final augmented dataset $\mathcal{D}_N$ is generated. Eventually, we train the final model $\hat{M}_{S \to T}$ on the dataset $\mathcal{D}_N$. For a clearer presentation, Algorithm 1 summarizes the process concretely. In the experiments, unless specified otherwise, we use the default setup of $k = 3$ and $N = 1$.

### 3.2 Relation with existing methods

Our method shares certain similarities with a variety of existing techniques, namely, data augmentation, back-translation, ensemble of models, knowledge distillation and multi-agent dual learning.

**Data augmentation** Our approach is genuinely a data augmentation method. Fadaee et al. [6] proposed an augmentation strategy which targets rare words to improve low-resource translation. Wang et al. [26] suggested to simply replace random words with other words in the vocabularies. Our approach is distinct from these methods in that it does not randomly corrupt the data and train the model on the augmented data on the fly. Instead, it transforms the data into synthetic translations, which follow different model distributions.

**Back-translation** Our method is similar to back-translation, which has been employed to generate synthetic data from target-side extra monolingual data. Sennrich et al. [17] were the first to propose such strategy, while Edunov et al. [4] refined it at scale. Our method's main advantage is that it does not require any extra monolingual data. Our technique also differs from previous work in that it additionally employs forward translation, which we have shown to be important (see §5.4).

**Ensemble of models**   Using multiple models to average the predictions and reduce variance is a typical feature of ensemble methods [16]. However, the drawback is that the testing parameters and computations are multiple times more than an individual model. While our diversification approach correlates with model ensembles (§5.1), it does not suffer this disadvantage.

**Knowledge distillation**   Knowledge distillation [13, 8] involves pre-training a big teacher model and using its predictions (forward translation) to train a smaller student as the final model. In comparison to that, our method additionally employs back-translation and involves multiple backward and forward "teachers". We use all backward, forward, as well as the original data to train the final model without any parameter reduction. We also repeat the process multiple times. In this context, our method also differs from the ensemble knowledge distillation method [7], which uses the teachers to jointly generate a single version of data. Our method on the other hand uses the teachers to individually generate various versions of synthetic data.

**Multi-agent dual learning**   Multi-agent dual learning [27] involves leveraging duality with multiple forward and backward agents. Similar to [7], this method combines multiple agents in an ensembling manner to form forward ($F_\alpha$) and backward ($G_\beta$) teachers. Then, it simultaneously optimizes the reconstruction losses $\Delta_x(x, G_\beta(F_\alpha(x)))$ and $\Delta_y(y, F_\alpha(G_\beta(y)))$ to train the final dual models. As a result, the two models are coupled and entangled. On the other hand, our method does not combine the agents in this way, nor does it optimize any reconstruction objective.

Our approach is also related but substantially different from the mixture of experts for diverse MT [20], iterative back-translation [11] and copied monolingual data for NMT [2]; see the Appendix for further details about these comparisons.

# 4   Experiments

In this section, we present experiments to demonstrate that our data diversification approach improves translation quality in many translation tasks, encompassing WMT and IWSLT tasks, and high- and low-resource translation tasks. Due to page limit, we describe the setup for each experiment briefly in the respective subsections and give more details in the Appendix.

## 4.1   WMT'14 English-German and English-French translation tasks

**Setup.**   We conduct experiments on the standard WMT'14 English-German (En-De) and English-French (En-Fr) translation tasks. The training datasets contain about 4.5M and 35M sentence pairs respectively. The sentences are encoded with joint Byte-Pair Encoding (BPE) [18] with 32K operations.We use newstest2013 as the development set, and newstest2014 for testing. Both tasks are considered high-resource tasks as the amount of parallel training data is relatively large. We use the Transformer [24] as our NMT model and follow the same configurations as suggested by Ott et al. [15]. When augmenting the datasets, we filter out the duplicate pairs, which results in training datasets of 27M and 136M pairs for En-De and En-Fr, respectively.[4] The data generation process costs approximately 30% the time to train the baseline. We average the last 5 checkpoints and we do not use any extra monolingual data.

**Results.**   From the results on WMT newstest2014 testset in Table 2, we observe that the scale Transformer [15], which originally gives 29.3 BLEU in the En-De task, now gives 30.7 BLEU with our data diversification strategy, setting a new SOTA. Our approach yields an improvement of 1.4 BLEU over the without-diversification model and 1.0 BLEU over the previous SOTA reported on this task by Wu et al. [28].[5] Our approach also outperforms other non-intrusive extensions, such as multi-agent dual learning and knowledge distillation by a good margin (0.7-3.1 BLEU). Similar observation can be drawn for WMT'14 En-Fr task. Our strategy establishes a new SOTA of 43.7 BLEU, exceeding the previous (reported) SOTA by 0.5 BLEU. It is important to mention that while our method increases the overall training time (including the time to train the base models), training a single Transformer model for the same amount of time only leads to overfitting.

Table 2: BLEU scores on newstest2014 for WMT'14 English-German (En-De) and English-French (En-Fr) translation tasks. Distill (T>S) (resp. T=S) indicates the teacher model is larger than (resp. equal to) the student model.

| Method | WMT'14 | |
| | En-De | En-Fr |
| --- | --- | --- |
| Transformer [24][†] | 28.4 | 41.8 |
| Trans+Rel. Pos [19][†] | 29.2 | 41.5 |
| Scale Transformer [15] | 29.3 | 42.7[6] |
| Dynamic Conv [28][†] | 29.7 | 43.2 |
| **Transformer with** | | |
| Multi-Agent [27][†] | 30.0 | - |
| Distill (T>S) [13] | 27.6 | 38.6 |
| Distill (T=S) [13] | 28.4 | 42.1 |
| Ens-Distill [7] | 28.9 | 42.5 |
| **Our Data Diversification with** | | |
| Scale Transformer [15] | **30.7** | **43.7** |

Table 3: BLEU scores on IWSLT'14 English-German (En-De), German-English (De-En), and IWSLT'13 English-French (En-Fr) and French-English (Fr-En) translation tasks. Superscript [†] denotes the numbers are reported from the paper, others are based on our runs.

| Method | IWSLT'14 | | IWSLT'13 | |
| | En-De | De-En | En-Fr | Fr-En |
| --- | --- | --- | --- | --- |
| **Baselines** | | | | |
| Transformer | 28.6 | 34.7 | 44.0 | 43.3 |
| Dynamic Conv | 28.7 | 35.0 | 43.8 | 43.5 |
| **Transformer with** | | | | |
| Multi-Agent[†] | 28.9 | 34.7 | - | - |
| Distill (T>S) | 28.0 | 33.6 | 43.4 | 42.9 |
| Distill (T=S) | 28.5 | 34.1 | 44.1 | 43.4 |
| Ens-Distill | 28.8 | 34.7 | 44.3 | 43.9 |
| **Our Data Diversification with** | | | | |
| Transformer | **30.6** | 37.0 | **45.5** | **45.0** |
| Dynamic Conv | **30.6** | **37.2** | 45.2 | 44.9 |

## 4.2 IWSLT translation tasks

**Setup.** We evaluate our approach in IWSLT'14 English-German (En-De) and German-English (De-En), IWSLT'13 English-French (En-Fr) and French-English (Fr-En) translation tasks. The IWSLT'14 En-De training set contains about 160K sentence pairs. We randomly sample 5% of the training data for validation and combine multiple test sets IWSLT14.TED.{dev2010, dev2012, tst2010, tst1011, tst2012} for testing. The IWSLT'13 En-Fr dataset has about 200K training sentence pairs. We use the IWSLT15.TED.tst2012 set for validation and the IWSLT15.TED.tst2013 set for testing. We use BPE for all four tasks. We compare our approach against two baselines that do not use our data diversification: Transformer [24] and Dynamic Convolution [28].

**Results.** From Table 3 we see that our method substantially and consistently boosts the performance in all the four translation tasks. In the En-De task, our method achieves up to 30.6 BLEU, which is 2 BLEU above the Transformer baseline. Similar trend can also be seen in the remaining De-En, En-Fr, Fr-en tasks. The results also show that our method is agnostic to model architecture, with both the Transformer and Dynamic Convolution achieving high gains. In contrary, other methods like knowledge distillation and multi-agent dual learning show minimal improvements on these tasks.

## 4.3 Low-resource translation tasks

Having demonstrated the effectiveness of our approach in high-resource languages like English, German and French, we now evaluate our approach performs on low-resource languages. For this, we use the English-Nepali and English-Sinhala low-resource setup proposed by Guzmán et al. [10]. Both Nepali (Ne) and Sinhala (Si) are challenging domains since the data sources are particularly scarce and the vocabularies and grammars are vastly different from high-resource language like English.

**Setup.** We evaluate our method on the supervised setup of the four low-resource translation tasks: En-Ne, Ne-En, En-Si, and Si-En. We compare our approach against the baseline in Guzmán et al. [10]. The English-Nepali and English-Sinhala parallel datasets contain about 500K and 400K sentence pairs respectively. We replicate the same setup as done by Guzmán et al. [10] and use their dev set for development and devtest set for testing. We use $k = 3$ in our data diversification experiments.
**Results.** From the results in Table 4, we can notice that our method consistently improves the performance by more than 1 BLEU in all four tested tasks. Specifically, the method achieves 5.7, 8.9, 2.2, and 8.2 BLEU for En-Ne, Ne-En, En-Si and Si-En tasks, respectively. In absolute terms, these are 1.4, 1.3, 2.2 and 1.5 BLEU improvements over the baseline model [10]. Without any monolingual data involved, our method establishes a new state of the art in all four low-resource tasks.

Table 4: Performances on low-resource translations. As done by Guzmán et al. [10], the from-English pairs are measured in tokenized BLEU, while to-English are measured in detokenized SacreBLEU.

| Method | En-Ne | Ne-En | En-Si | Si-En |
|---|---|---|---|---|
| Guzmán et al. [10] | 4.3 | 7.6 | 1.0 | 6.7 |
| Data Diversification | **5.7** | **8.9** | **2.2** | **8.2** |

Table 5: Diversification preserves the effects of ensembling, but does not change $|\Theta|$ and flops.

| | $|\Theta|$ | IWSLT'14 | | IWSLT'13 | | WMT |
|---|---|---|---|---|---|---|
| | flops | En-De | De-En | En-Fr | Fr-En | En-De |
| Baseline | 1x | 28.6 | 34.7 | 44.0 | 43.3 | 29.3 |
| Ensemble | **7x** | 30.2 | 36.5 | **45.5** | 44.9 | 30.3 |
| Ours | 1x | **30.6** | **37.0** | **45.5** | **45.0** | **30.7** |

Table 6: BLEU scores for forward and backward diversification in comparison to bidirectional diversification and the baseline on IWSLT'14 En-De and De-En tasks.

| Task | Baseline | Backward | Forward | Bidirectional |
|---|---|---|---|---|
| En-De | 28.6 | 29.2 | 29.86 | 30.6 |
| De-En | 34.7 | 35.8 | 35.94 | 37.0 |

# 5 Understanding data diversification

We propose several logical hypotheses to explain why and how data diversification works as well as provide a deeper insight to its mechanism. We conduct a series of experimental analysis to confirm or reject such hypotheses. As a result, certain hypotheses are confirmed by the experiments, while some others, though being intuitive, are experimentally rejected. In this section, we explain the hypotheses that are empirically verified, while we elaborate the failed hypotheses in the Appendix.

## 5.1 Ensemble effects

**Hypothesis** *Data diversification exhibits a strong correlation with ensemble of models.*

**Experiments** To show this, we perform inference with an ensemble of seven (7) models and compare its performance with ours. We evaluate this setup on the WMT'14 En-De, and IWSLT'14 En-De, De-En, IWSLT'13 En-Fr and Fr-En translation tasks. The results are reported in Table 5. We notice that the ensemble of models outdoes the single-model baseline by 1.3 BLEU in WMT'14 and 1.0-2.0 BLEU in IWSLT tasks. These results are particularly comparable to those achieved by our technique. This suggests that our method may exhibit an ensembling effect. However, note that an ensemble of models has a major drawback that it requires $N$ (7 in this case) times more computations and parameters to perform inference. In contrary, our method does not have this disadvantage.

**Explanation** Intuitively, different models (initialized with different random seeds) trained on the original dataset converge to different local optima. As such, individual models tend to have high variance. Ensembles of models are known to help reduce variance, thus improves the performance. Formally, suppose a single-model $M_i \in \{M_1, ..., M_N\}$ estimates a model distribution $p_{M_i}$, which is close to the data generating distribution $p_{data}$. An ensemble of models averages multiple $p_{M_i}$ (for $i = 1, \ldots, N$), which leads to a model distribution that is closer to $p_{data}$ and improves generalization.

Our strategy may achieve the same effect by forcing a single-model $\hat{M}$ to learn from the original data distribution $p_{data}$ as well as multiple synthetic distributions $\mathcal{D}'_i \sim p_{M_i}$ for $i = 1, \ldots, N$, simultaneously. Following Jensen's Inequality [12], our method optimizes the upper bound:

$$\mathbb{E}_{Y \sim U(M_1(X),...,M_N(X)), X \sim p_{data}} \log p_{\hat{M}}(Y|X) \leq \sum_j \log[\frac{1}{N} \sum_i^N p_{\hat{M}}(y_j^i|y_{<j}^i, X)] \quad (2)$$

where $U$ is uniform sampling, $y_j^i = \text{argmax}_{y_j} p_{M_i}(y_j|y_{<j}, X)$. Let $\max_{y_j^k} \frac{1}{N} \sum_i^N p_{M_i}(y_j^k|y_{<j}, X)$ be the token-level probability of an ensemble of models $M_i$ and $V$ be the vocabulary. Experimentally, we observe that the final model $\hat{M}$ tends to outperform when the following condition is met:

$$\mathbb{E}_{X \sim p_{data}} \left[ \frac{1}{N} \sum_i^N p_{\hat{M}}(y_j^i|y_{<j}^i, X) \right] \leq \mathbb{E}_{X \sim p_{data}} \left[ \max_{y_j^k} \frac{1}{N} \sum_i^N p_{M_i}(y_j^k|y_{<j}, X) \right] \text{ with } y_j^k \in V \quad (3)$$

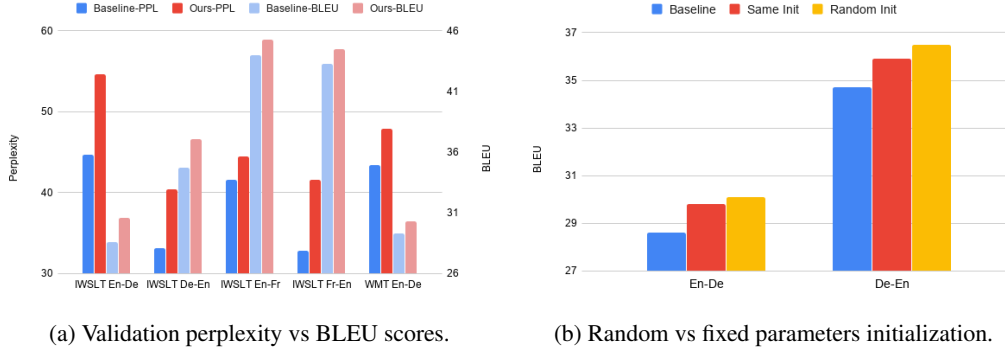

(a) Validation perplexity vs BLEU scores.　　　　(b) Random vs fixed parameters initialization.

Figure 1: Relationship between validation perplexity vs the BLEU scores (1a) and the effects of random initialization (1b) in the IWSLT En-De, De-En, En-Fr, Fr-En and WMT'14 En-De tasks.

Condition 3 can be met naturally at the beginning of the training process, but is not guaranteed at the end. We provide further analysis and supporting experiments in the Appendix.

## 5.2 Perplexity vs. BLEU score

**Hypothesis** *Data diversification sacrifices perplexity for better BLEU score.*

**Experiments** We tested this hypothesis as follows. We recorded the validation perplexity when the models fully converge for the baseline setup and for our data diversification method. We report the results in Figure 1a for WMT'14 En-De, IWSLT'14 En-De, De-En, IWSLT'13 En-Fr and Fr-En tasks. The left axis of the figure shows Perplexity (PPL) values for the models, which compares the *dark* blue (baseline) and red (our) bars. Meanwhile, the right axis shows the respective BLEU scores for the models as reflected by the *faded* bars.

**Explanation** Common wisdom tells that the lower perplexity often leads to better BLEU scores. In fact, our NMT models are trained to minimize perplexity (equivalently, cross entropy loss). However, existing research [23] also suggests that sometimes sacrificing perplexity may result in better generalization and performance. As shown in Figure 1a, our models consistently show higher perplexity compared to the baseline in all the tasks, though we did not have intention to do so. As a result, the BLEU score is also consistently higher than the baseline.

## 5.3 Initial parameters vs. diversity

**Hypothesis** *Models with different initial parameters increase diversity in the augmented data, while the ones with fixed initial parameters decrease it.*

**Experiments and Explanation** With the intuition that diversity in training data improves translation quality, we speculated that the initialization of model parameters plays a crucial role in data diversification. Since neural networks are susceptible to initialization, it is possible that different initial parameters may lead the models to different convergence paths [9] and thus different model distributions, while models with the same initialization are more likely to converge in similar paths. To verify this, we did an experiment with initializing all the constituent models ($M_{S \to T,n}^i$, $M_{T \to S,n}^i$, $\hat{M}_{S \to T}$) with the same initial parameters to suppress data diversity. We conducted this experiment on the IWSLT'14 English-German and German-English tasks. We used a diversification factor of $k = 1$ only in this case. The results are shown in Figure 1b. Apparently, the BLEU scores of the fixed (same) initialization drop compared to the randomized counterpart in both language pairs. However, its performance is still significantly higher than the single-model baseline. This suggests that initialization is not the only contributing factor to diversity. Indeed, even though we are using the same initial checkpoint, each constituent model is trained on a different dataset and and learns to estimate a different distribution.

## 5.4 Forward-translation is important

**Hypothesis** *Forward-translation is as vital as back-translation.*

**Experiments and Explanation**  We separate our method into forward and backward diversification, in which we only train the final model ($\hat{M}_{S \to T}$) with the original data augmented by either the translations of the forward models ($M^i_{S \to T, n}$) or those of the backward models ($M^i_{T \to S, n}$) separately. We compare those variants with the bidirectionally diversified model and the single-model baseline. As shown in Table 6, the forward and backward methods perform worse than the bidirectional counterpart but still better than the baseline. However, it is worth noting that diversification with forward models outperforms the one with backward models, as recent research has focused mainly on back-translation where only backward models are used [17, 4]. Our finding is similar to Zhang and Zong [29], where the authors used source-side monolingual data to improve BLEU score.

## 6   Unaffected by the translationese effect

Data diversification is built on the foundation of back-translation [17, 4]. However, in a recent work, Edunov et al. [5] point out that back-translation suffers from the *translationese effect* [25], where back-translation only improves the performance when the source sentences are translationese but does not offer any improvement when the sentences are natural text.[7] In other words, back-translation can be ineffective in practice because our goal is to translate natural text, not translated text. Thus, it is important to test whether our data diversification method also suffers from this effect.

To verify this, we measure the BLEU improvements of data diversification over the baseline [15] in the WMT'14 English-German setup (§4.1) in all 3 scenarios laid out by Edunov et al. [5]: (*i*) natural source $\to$ translationese target ($X \to Y^*$), (*ii*) translationese source $\to$ natural target ($X^* \to Y$), and (*iii*) translationese of translationese of source to translationese of target ($X^{**} \to Y^*$). We use the same test sets provided by Edunov et al. [5] to conduct this experiment.[8]

As shown in Table 7, out method consistently outperforms the baseline in all three scenarios while Edunov et al. [5] show that back-translation [17] improves only in the $X^* \to Y$ scenario. Thus, our method is not effected by the translationese effect. Our explanation for this is that the mentioned back-translation technique [17] is a *semi-supervised* setup that uses extra natural monolingual data in the target. In our method, however, back-translation is conducted on the translationese part (target side) of the parallel data, and does not enjoy the introduction of extra natural text, which only benefits the $X^* \to Y$ scenario. Otherwise speaking, back-translation *with* and *without* monolingual data are two distinct setups that should not be confused or identically interpreted.

Table 7: BLEU evaluation of our method and the baseline on the translationese effect [5], in the WMT'14 English-German setup.

| WMT'14 En-De | $X \to Y^*$ | $X^* \to Y$ | $X^{**} \to Y^*$ |
|---|---|---|---|
| Baseline [15] | 31.35 | 28.47 | 38.59 |
| Our method | 33.47 | 30.38 | 41.03 |

## 7   Study on hyperparameters and back-translation

**Effect of different $k$ and $N$**  We first conduct experiments with the two hyper-parameters in our method – the diversification factor $k$ and the number of rounds $N$, to investigate how they affect the performance. Particularly, we test the effect of different $N$ on the IWSLT'14 En-De and De-En tasks, while the effect of different $k$ is tested on the WMT'14 En-De task. As shown in Table 8, increasing $N$ improves the performance but the gain margin is insignificant. Meanwhile, Table 9 shows that increasing $k$ significantly improves the performance until a specific saturation point. Note that increasing $N$ costs more than increasing $k$ while its gain may not be significant.

**Complementary to back-translation**  Our method is also complementary to back-translation (BT) [17]. To demonstrate this, we conducted experiments on the IWSLT'14 En-De and De-En tasks with

Table 8: BLEU scores for different rounds $N$

| IWSLT'14 | $N=1$ | $N=2$ | $N=3$ |
|---|---|---|---|
| En-De | 30.4 | 30.6 | 30.6 |
| De-En | 36.8 | 36.9 | 37.0 |

Table 9: BLEU scores for different factors $k$.

| WMT'14 | $k=1$ | $k=2$ | $k=3$ | $k=4$ | $k=5$ | $k=6$ |
|---|---|---|---|---|---|---|
| En-De | 29.8 | 30.1 | 30.7 | 30.7 | 30.7 | 30.6 |

extra monolingual data extracted from the WMT'14 En-De corpus. In addition, we also compare our method against the back-translation baseline in the WMT'14 En-De task with extra monolingual data from News Crawl 2009. We use the *big* Transformer as the final model in all our back-translation experiments. Further details of these experiments are provided in the Appendix. As reported in Table 10, using back-translation improves the baseline performance significantly. However, using our data diversification strategy with such monolingual data boosts the performance further with additional +1.0 BLEU over the back-translation baselines.

Table 10: BLEU scores for models with and without back-translation (BT) on the IWSLT'14 English-German (En-De), German-English (De-En) and WMT'14 En-De tasks. Column $|D|$ shows the total data used in back-translation compared to the original parallel data.

| Task | No back-translation | | With back-translation | | |
|---|---|---|---|---|---|
| | Baseline | Ours | $|D|$ | Baseline | Ours |
| IWSLT'14 En-De | 28.6 | 30.6 | $29\times$ | 30.0 | **31.8** |
| IWSLT'14 De-En | 34.7 | 37.0 | $29\times$ | 37.1 | **38.5** |
| WMT'14 En-De | 29.3 | 30.7 | $2.4\times$ | 30.8 | **31.8** |

## 8   Conclusion

We have proposed a simple yet effective method to improve translation performance in many standard machine translation tasks. The approach achieves state-of-the-art in the WMT'14 English-German translation task with 30.7 BLEU. It also improves in IWSLT'14 English-German, German-English, IWSLT'13 English-French and French-English tasks by 1.0-2.0 BLEU. Furthermore, it outperforms the baselines in the low-resource tasks: English-Nepali, Nepali-English, English-Sinhala, Sinhala-English. Our experimental analysis reveals that our approach exhibits a strong correlation with ensembles of models. It also trades perplexity off for better BLEU score. We have also shown that the method is complementary to back-translation with extra monolingual data as it improves the back-translation performance significantly.

## Broader Impact

Our work has a potential positive impact on the application of machine translation in a variety of languages. It helps boost performance in both high- and low-resource languages. The method is simple to implement and applicable to virtually all existing machine translation systems. Future commercial and humanitarian translation services can benefit from our work and bring knowledge of one language to another, especially for uncommon language speakers such as Nepalese and Sri Lankan. On the other hand, our work needs to train multiple models, which requires more computational power or longer time to train the final model.

## Acknowledgement

We are grateful to our anonymous reviewers and meta-reviewer for their feedback and comments, which help us improve our paper. We thank Thanh-Tung Nguyen for our discussions with him about the ideas. We also thank Prathyusha Jwalapuram for proofreading the paper. Xuan-Phi is supported by the A*STAR Computing and Information Science (ACIS) scholarship, provided by the Agency for Science, Technology and Research Singapore (A*STAR). Shafiq Joty would like to thank the funding support from NRF (NRF2016IDM-TRANS001-062), Singapore.

## Footnotes

[1]Code: https://github.com/nxphi47/data_diversification

[2]As of submission deadline, we report SOTA in the standard WMT'14 setup without monolingual data.

[3]Parameters $|\Theta|$ do not increase as we can discard the intermediate models after using them.

[4]There were 14% and 22% duplicates, respectively. We provide extra diversity analysis in the Appendix.

[5]We could not reproduce the results reported by Wu et al. [28] using their code. We only achieved 29.2 BLEU for this baseline, while our method applied to it gives 30.1 BLEU.

[6]Ott et al. [15] reported 43.2 BLEU in En-Fr. However, we could achieve only 42.7 using their code, based on which our data diversification gives 1.0 BLEU gain.

[7]Translationese refers to the unique characteristics of translated text (e.g., simplification, explicitation, normalization) compared to text originally written in a given language. This happens because when translating a text, translators usually have to make trade-offs between fidelity (to the source) and fluency (in the target).

[8]Note that our setup uses the WMT'14 training set while Edunov et al. [5] use the WMT'18 training set.

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
