[Supplementary Material]

# Appendix for Data Diversification: A Simple Strategy For Neural Machine Translation

**Xuan-Phi Nguyen**[1,3]**, Shafiq Joty**[1,2]**, Wu Kui**[3]**, Ai Ti Aw**[3]
[1]Nanyang Technological University
[2]Salesforce Research
[3]Institute for Infocomm Research (I[2]R), A*STAR
Singapore
{nguyenxu002@e.ntu,srjoty@ntu}.edu.sg
{wuk,aaiti}@i2r.a-star.edu.sg

In the following supplementary materials, we discuss the other hypotheses that are not supported by the experiments. After that, we present mathematically our data diversification method is correlated to ensembles of models. Finally, we describe the training setup for our back-translation experiments.

## A    Further comparison

We continue to differentiate our method from other existing works. First, Shen et al. [12] also seek to generate diverse set of translations using mixture of experts, not to improve translation quality like ours. In this method, multiple experts are tied into a single NMT model to be trained to generate diverse translations through EM optimization. It does not employ data augmentation, neither forward nor backward translations. Our method does not train multiple peer models with EM training either.

Second, iterative back-translation [4] employs back-translation to augment the data in multiple rounds. In each round, a forward (or backward) model takes turn to play the "back-translation" role to train the backward (or forward) model. The role is switched in the next round. In our approach, both directions are involved in each round and multiple models to the achieve ensembling effect.

Third, Currey et al. [1] propose to generate bitext from the target-side monolingual data by just copying the target sentence into the source sentence. In other words, source and target are identical. Our method does not use copying practice nor any extra monolingual data.

## B    Experimental setup details

**WMT'14 setup.**    We conduct experiments on the WMT'14 English-German (En-De) and English-French (En-Fr) translation tasks. The training datasets contain about 4.5 million and 35 million sentence pairs respectively. The sentences are encoded with Byte-Pair Encoding (BPE) [11] with 32,000 operations, which results in a shared-vocabulary of 32,768 tokens for En-De and 45,000 tokens for En-Fr. We use newstest2013 as the development set, and newstest2014 for testing. Both tasks are considered as high-resource tasks as the amount of parallel training data is relatively large.

We use the Transformer [13] as our NMT model and follow the same configurations as suggested by Ott et al. [8]. The model has 6 layers, each of which has model dimension $d_{model} = 1024$, feed-forward dimension $d_{ffn} = 4096$, and 16 attention heads. Adam optimizer [6] was used with the similar learning rate schedule as Ott et al. [8] — 0.001 learning rate, 4,000 warm-up steps, and a batch size of 450K tokens. We use a dropout rate of 0.3 for En-De and 0.1 for En-Fr. We train the models for 45,000 updates. The data generation process costs 30% the time to train the baseline.

For data diversification, we use a diversification factor $k = 3$ for En-De and $k = 2$ for En-Fr. When augmenting the datasets, we filter out the duplicate pairs, which results in training datasets of 27M

and 136M sentence pairs for En-De and En-Fr, respectively.[1] Note that we do not use any extra monolingual data. For inference, we average the last 5 checkpoints of the final model and use a beam size of 5 and a length penalty of 0.6. We measure the performance in standard tokenized BLEU.

**IWSLT setup.** We also show the effectiveness of our approach in IWSLT'14 English-German (En-De) and German-English (De-En), IWSLT'13 English-French (En-Fr) and French-English (Fr-En) translation tasks. The IWSLT'14 En-De training set contains about 160K sentence pairs. We randomly sample 5% of the training data for validation and combine multiple test sets IWSLT14.TED.{dev2010, dev2012, tst2010, tst1011, tst2012} for testing. The IWSLT'13 En-Fr dataset has about 200K training sentence pairs. We use the IWSLT15.TED.tst2012 set for validation and the IWSLT15.TED.tst2013 set for testing. We use BPE for all four tasks. This results in a shared vocabulary of 10,000 tokens for English-German pair and 32,000 tokens for English-French pair.

We compare our approach against two baselines that do not use our data diversification: Transformer [13] and Dynamic Convolution [14]. In order to make a fair comparison, for the baselines and our approach, we use the **base** setup of the Transformer model. Specifically, the models have 6 layers, each with model dimensions $d_{model} = 512$, feed-forward dimensions $d_{ffn} = 1024$, and 4 attention heads. We use a dropout of $0.3$ for all our IWSLT experiments. The models are trained for 500K updates and selected based on the validation loss. Note that we do not perform checkpoint averaging for these tasks, rather we run the experiments for 5 times with different random seeds and report the mean BLEU scores to provide more consistent and stable results. For inference, we use a beam size of 5, a length penalty of $1.0$ for En-De, $0.2$ for En-Fr, and $2.5$ for Fr-En pair.

**Low-resource setup.** We evaluate our data diversification strategy on the supervised setups of the four low-resource translation tasks: En-Ne, Ne-En, En-Si, and Si-En. We compare our approach against the baseline proposed in [3]. The English-Nepali parallel dataset contains about 500K sentence pairs, while the English-Sinhala dataset has about 400K pairs. We use the provided dev set for development and devtest set for testing.

In terms of training parameters, we replicate the same setup as done by Guzmán et al. [3]. Specifically, we use the base Transformer model with 5 layers, each of which has 2 attention heads, $d_{model} = 512$, $d_{ffn} = 2048$. We use a dropout rate of $0.4$, label smoothing of $0.2$, weight decay of $10^{-4}$. We train the models for 100 epochs with batch size of 16,000 tokens. We select the inference models and length penalty based on the validation loss. The Nepali and Sinhala corpora are tokenized using the Indic NLP library.[2] We reuse the provided shared vocabulary of 5000 tokens built by BPE learned with the *sentencepiece* library.[3]

For inference, we use beam search with a beam size of 5, and a length penalty of $1.2$ for Ne-En and Si-En tasks, $0.9$ for En-Ne and $0.5$ for En-Si. We report tokenized BLEU for from-English tasks and detokenized SacredBLEU [9] for to-English tasks. We use $k = 3$ in our data diversification experiments.

## C   Diversity analysis

As mentioned, by training multiple models with different random seeds, the generated translations from the training set yield only 14% and 22% duplicates for En-De and En-Fr, respectively. These results may be surprising as we might expect more duplicates. Therefore, we performed further diversity analysis.

To evaluate the diversity the teacher models brought in data diversification, we compare them using the BLEU/Pairwise-BLEU benchmark proposed by Shen et al. [12]. This benchmark measures the diversity and quality of multiple hypotheses, generated by multiple models or a mixture of experts, given a source sentence. Specifically, we use our forward models trained on WMT'14 English-German and English-French and measure the BLEU and Pairwise-BLEU scores in the provided test set [12]. The results are reported in Table 1, where we compare the diversity and quality of our method against the mixture of experts method provided by Shen et al. [12] (hMup) and other

Table 1: WMT'14 English-German (En-De) and English-French (En-Fr) diversity performances in BLEU and Pairwise-BLEU scores, tested on the test set provided by Shen et al. [12]. Lower Pairwise-BLEU means more diversity, higher BLEU means better quality.

| Method | Pairwise-BLEU | | BLEU | |
|---|---|---|---|---|
| | En-De | En-Fr | En-De | En-Fr |
| Sampling | 24.1 | 32.0 | 37.8 | 46.5 |
| Beam | 73.0 | 77.1 | 69.9 | 79.8 |
| Div-beam | 53.7 | 64.9 | 60.0 | 72.5 |
| hMup | 50.2 | 64.0 | 63.8 | 74.6 |
| Human | 35.5 | 46.5 | 69.6 | 76.9 |
| Ours | 57.1 | 70.1 | 69.5 | 77.0 |

Table 2: Improvements of data diversification under conditions with- and without- dropout in the IWSLT'14 English-German and German-English.

| Task | Baseline | Ours | Gain |
|---|---|---|---|
| **Dropout = 0.3** | | | |
| En-De | 28.6 | 30.1 | +1.5 (5%) |
| De-En | 34.7 | 36.5 | +1.8 (5%) |
| **Dropout = 0.0** | | | |
| En-De | 25.7 | 27.5 | +1.8 (6%) |
| De-En | 30.7 | 32.5 | +1.8 (5%) |

baselines, such as sampling [2], diverse bearm search [7]. All methods, ours and the baselines, use the big Transformer model.

As it can be seen in En-De experiments, our method is less diverse than the mixture of experts (hMup) [12] and diverse beam search [7] (57.1 versus 50.2 and 53.7 Pairwise-BLEU). However, translations of our method are of better quality (69.5 BLEU), which is very close to human performance. Meanwhile, our method achieve similar quality to beam, but yields better diversity than this approach. The same conclusion can be derived from the WMT'14 English-French experiments.

## D  Failed Hypotheses

In addition to the successful hypotheses that we described in the paper, we speculated other possible hypotheses that were eventually not supported by the experiments despite being intuitive. We present them in this section for better understanding of the approach.

**Effects of Dropout.**   First, given that parameter initialization affects diversity, it is logical to assume that **Dropout** will magnify the diversification effects. In other words, we expected that removing dropout would result in less performance boost offered by our method than when dropout is enabled. However, our empirical results did not support this.

We ran experiments to test whether non-zero dropout magnify the improvements of our method over the baseline. We trained the single-model baseline and our data diversification's teacher models (both backward, forward models) as well as the student model in cases of dropout $= 0.3$ and dropout $= 0.0$ in the IWSLT'14 English-German and German-English tasks. We used factor $k = 1$ in these experiments. As reported in Table 2, the no-dropout versions perform much worse than the non-zero dropout versions in all experiments. However, the gains made by our data diversification with dropout are not particularly higher than the non-dropout counterpart. This suggests that dropout may not contribute to the diversity of the synthetic data.

Table 3: Improvements of data diversification under conditions maximum likelihood and beam search in the IWSLT'14 English-German and German-English.

| Task | Baseline | Ours | |
|---|---|---|---|
| | | Beam=1 | Beam=5 |
| En-De | 28.6 | 30.3 | 30.4 |
| De-En | 34.7 | 36.6 | 36.8 |

**Effects of Beam Search.** We hypothesized that beam search would generate more diverse synthetic translations of the original dataset, thus increases the diversity and improves generalization. We tested this hypothesis by using greedy decoding (beam size = 1) to generate the synthetic data and compare its performance against beam search (beam size = 5) counterparts. We again used the IWSLT'14 English-German and German-English as a testbed. Note that for testing with the final model, we used the same beam search (beam size = 5) procedure for both cases. As shown in Table 3, the performance of greedy decoding is not particularly reduced compared to the beam search versions.

## E   Correlation with Ensembling

In this section, we show that our data diversification method is optimizing its model distribution to be close to the ensemble distribution under the condition that the final model is randomly initialized. We also show that such condition can be easily met in our experiments. Specifically, let the constituent models of an ensemble of models be $M_i \in \{M_1, ..., M_N\}$ and each model $M_i$ produces a model distribution probability $p_{M_i}(Y|X)$ (for $i = 1, \ldots, N$) of the true probability $P(Y|X)$, $X$ and $Y = (y_1, ..., y_m)$ be the source and target sentences sampled from the data generating distribution $p_{data}$, $\hat{M}$ be the final model in our data diversification strategy. In addition, without loss of generality, we assume that the final model $\hat{M}$ is only trained on the data generated by forward models $M_{S \to T, r}^i$. We also have token-level probability $P(y_j|y_{<j}, X)$ such that:

$$\log P(Y|X) = \sum_j \log P(y_j|y_{<j}, X) \tag{1}$$

For the sake of brevity, we omit $X$ in $P(y_j|y_{<j}, X)$ and use only $P(y_j|y_{<j})$ in the remaining of description. With this, our data diversification method maximizes the following expectation:

$$
\begin{aligned}
\mathbb{E}_{Y \sim U(M_1(X),...,M_N(X)), X \sim p_{data}} \log p_{\hat{M}}(Y|X) &= \mathbb{E}_{Y \sim Y_c} \sum_j \log p_{\hat{M}}(y_j|y_{<j}) \\
&= \sum_{Y \sim Y_c} \frac{1}{N} \sum_j \log p_{\hat{M}}(y_j|y_{<j}) \\
&= \sum_j \frac{1}{N} \sum_{Y \sim Y_c} \log p_{\hat{M}}(y_j|y_{<j})
\end{aligned}
\tag{2}
$$

where $U$ denotes uniform sampling, $Y_c = U(M_1(X), ..., M_N(X))$. According to Jensen's Inequality [5], we have:

$$\sum_j \frac{1}{N} \sum_{Y \sim Y_c} \log p_{\hat{M}}(y_j|y_{<j}) \le \sum_j \log[\frac{1}{N} \sum_{Y \sim Y_c} p_{\hat{M}}(y_j|y_{<j})] = \sum_j \log[\frac{1}{N} \sum_i^N p_{\hat{M}}(y_j^i|y_{<j}^i)] \tag{3}$$

where $y_j^i = \text{argmax}_{y_j} p_{M_i}(y_j|y_{<j})$. Meanwhile, we define $\max_{y_j^k} \frac{1}{N} \sum_i^N p_{M_i}(y_j^k)$ as the averaged probability for output token $y_j^e = \text{argmax}_{y_j} \frac{1}{N} \sum_i^N p_{M_i}(y_j)$ of an ensemble of models. Then, since the maximum function is convex, we have the following inequality:

$$\mathbb{E}\Big[ \max_{y_j^k} \frac{1}{N} \sum_i^N p_{M_i}(y_j^k|y_{<j}) \Big] \le \mathbb{E}\Big[ \frac{1}{N} \sum_i^N \max_{y_j^k} p_{M_i}(y_j^k|y_{<j}^i) \Big] = \mathbb{E}\Big[ \frac{1}{N} \sum_i^N p_{M_i}(y_j^i|y_{<j}^i) \Big] \tag{4}$$

where $y_j^k \in V$ and $V$ is the vocabulary. In addition, with the above notations, by maximizing the expectation 2, we expect our method to automatically push the final model distribution $p_{\hat{M}}$, with

the term $\frac{1}{N}\sum_i^N p_{\hat{M}}(y_j^i|y_{<j}^i)$ close to the average model distribution of $\frac{1}{N}\sum_i^N p_{M_i}(y_j^i|y_{<j}^i)$ (the right-hand side of Eqn. 4).

Through experiments we will discuss later, we observe that our method is able to achieve high performance gain under the following conditions:

- Both sides of Eqn. 4 are tight, meaning they are almost equal. This can be realized when the teacher models are well-trained from the parallel data.

- The following inequality needs to be maintained and the training process should stop when the inequality no longer holds:

$$\mathbb{E}\Big[\frac{1}{N}\sum_i^N p_{\hat{M}}(y_j^i|y_{<j}^i)\Big] \leq \mathbb{E}\Big[\max_{y_j^k}\frac{1}{N}\sum_i^N p_{M_i}(y_j^k|y_{<j})\Big] \leq \mathbb{E}\Big[\frac{1}{N}\sum_i^N p_{M_i}(y_j^i|y_{<j}^i)\Big] \leq 1 \quad (5)$$

The left-side equality of condition 5 happens when all $y_j^i$ are identical $\forall i \in \{1,...,N\}$ and $y_j^i = y_j^e \forall y_j$. In the scenario of our experiments, condition 5 is easy to be met.

First, when the constituent models $M_i$ are well-trained (but not overfitted), the confidence of the models is also high. This results in the expectation $\mathbb{E}\Big[\max_{y_j^k}\frac{1}{N}\sum_i^N p_{M_i}(y_j^k|y_{<j})\Big]$ and $\mathbb{E}\Big[\frac{1}{N}\sum_i^N p_{M_i}(y_j^i|y_{<j}^i)\Big]$ comparably close to 1.0, compared to uniform probability of $1/|V|$ with $V$ is the target vocabulary and $|V| \gg N$. To test this condition, we empirically compute the average values of both terms over the IWSLT'14 English-German and IWSLT'13 English-French tasks. As reported in Table 4, the average probability $\mathbb{E}\Big[\max_{y_j^k}\frac{1}{N}\sum_i^N p_{M_i}(y_j^k|y_{<j})\Big]$ is around 0.74-0.77, while the average probability $\mathbb{E}\Big[\frac{1}{N}\sum_i^N p_{M_i}(y_j^i|y_{<j}^i)\Big]$ is always higher but close to the former term (0.76-0.79). Both of these terms are much larger than $1/|V| = 3 \times 10^{-5}$.

Second, as model $\hat{M}$ is randomly initialized, it is logical that $\mathbb{E}\frac{1}{N}\sum_i^N p_{\hat{M}}(y_j^i|y_{<j}^i) \approx 1/|V|$. Thus, under our experimental setup, it is likely that condition 5 is met. However, the condition can be broken when the final model $\hat{M}$ is trained until it overfits on the augmented data. This results in $\frac{1}{N}\sum_i^N p_{\hat{M}}(y_j^i|y_{<j}^i) \leq \mathbb{E}\Big[\max_{y_j^k}\frac{1}{N}\sum_i^N p_{M_i}(y_j^k)\Big]$. This scenario is possible because the model $\hat{M}$ is trained on many series of tokens $y_j^i$ with absolute confidence of 1, despite the fact that model $M_i$ produces $y_j^i$ with a relative confidence $p_{M_i}(y_j^i|y_{<j}^i) < 1$. In other words, our method does not restrict the confidence of the final model $\hat{M}$ for the synthetic data up to the confidence of their teachers $M_i$. Breaking this condition may cause a performance drop. As demonstrated in Table 4, the performances significantly drop as we overfit the diversified model $\hat{M}$ so that it is more confident in the predictions of $M_i$ than $M_i$ themselves (*i.e.,* 0.82 versus 0.76 probability for En-De). Therefore, it is recommended to maintain a final model's confidence on the synthetic data as high and close to, but not higher than the teacher models' confidence.

## F  Details on Back-translation experiments

In this section, we describe the complete training setup for the back-translation experiments presented in Section 6. First, for the back-translation baselines [10], we train the backward models following the same setup for the baseline Transformer presented in Section 4.1 for the WMT'14 En-De experiment and Section 4.2 for IWSLT'14 De-En and En-De experiments. For IWSLT experiments, we use the target-side corpora (En and De) from the WMT'14 English-German dataset to augment the IWSLT De-En and En-De tasks. We use the BPE code built from the original parallel data to transform these monolingual corpora into BPE subwords. This results in a total dataset of 4.66M sentence pairs, which is 29 times larger than the original IWSLT datasets. For the final model trained on the augmented data, we use the big Transformer with the same hyper-parameters as described in Section 4.1. However, note that we use the same shared vocabulary from the baseline setup for the back-translation experiments. On the other hand, for the WMT'14 English-German experiment, we use the German corpus derived from News Crawl 2009, which contains 6.4M sentences. Similarly, we use the same BPE code and shared vocabulary built from the parallel data to transform and encode

Table 4: The average value of $\mathbb{E}\left[\frac{1}{N}\sum_i^N p_{M_i}(y_j^i|y_{<j}^i)\right]$, $\mathbb{E}\left[\max_{y_j^k}\frac{1}{N}\sum_i^N p_{M_i}(y_j^k|y_{<j})\right]$ and $\mathbb{E}\left[\frac{1}{N}\sum_i^N p_{\hat{M}}(y_j^i|y_{<j}^i)\right]$ in the IWSLT'14 English-German, German-English, IWSLT'13 English-French and French-English tasks.

| | En-De | De-En | En-Fr | Fr-En |
|---|---|---|---|---|
| **Teacher models** $M_i$ | | | | |
| $\mathbb{E}\left[\frac{1}{N}\sum_i^N p_{M_i}(y_j^i|y_{<j}^i)\right]$ | 0.76 | 0.78 | 0.76 | 0.79 |
| $\mathbb{E}\left[\max_{y_j^k}\frac{1}{N}\sum_i^N p_{M_i}(y_j^k|y_{<j})\right]$ | 0.75 | 0.76 | 0.74 | 0.77 |
| Test BLEU | 28.6 | 34.7 | 44.0 | 43.3 |
| **Diversified Model** $\hat{M}$ | | | | |
| $\mathbb{E}\left[\frac{1}{N}\sum_i^N p_{\hat{M}}(y_j^i|y_{<j}^i)\right]$ | 0.74 | 0.74 | 0.73 | 0.75 |
| Test BLEU | 30.6 | 37.0 | 45.5 | 45.0 |
| **Overfitted Diversified Model** $\hat{M}$ | | | | |
| $\mathbb{E}\left[\frac{1}{N}\sum_i^N p_{\hat{M}}(y_j^i|y_{<j}^i)\right]$ | 0.82 | 0.86 | 0.84 | 0.89 |
| Test BLEU | 28.6 | 34.8 | 43.8 | 43.3 |

this monolingual corpus. The process produces a total dataset of 10.9M sentence pairs, which is 2.4 times larger than the original dataset. We use the big Transformer setup for all the WMT experiments.

Second, in terms of back-translation models trained with our data diversification strategy, we use the existing $k$ backward models to generate $k$ diverse sets of source sentences from the provided monolingual data. After that, we combine these datasets with the diverse dataset built by our method from the original parallel data. Then, we train the final model on this dataset. In addition, model setups and training parameters are identical to those used for the back-translation baselines.

## Footnotes

[1]There were 14% and 22% duplicates, respectively. We provide a diversity analysis in the Appendix.

[2]https://github.com/anoopkunchukuttan/indic_nlp_library

[3]https://github.com/google/sentencepiece