[Reviews · NeurIPS 2020]

Review 1

Summary and Contributions: This paper investigates data diversification for neural machine translation. The authors augment data from parallel corpus (x, y), which is training data, and build forward translation data (x, y') and back-translation data (x', y) by utilizing source side data x and target side data y. The proposed method achieved the state-of-the-art performance in WMT'14 English-German and English-French translation task, and substantially improves on 8 other translation tasks.

Strengths: The authors showed that even parallel corpora can be useful to augment data, since the existing research work discussed monolingual usage. The idea is straightforward, and the proposed method gives +1-2 BLEU score improvement on many language translation tasks. Analysis parts shows some evidence of why the proposed approach works.

Weaknesses: - In Table 2, what about the performance of vanilla Transformer with the proposed approach? It's clearer to report the baseline + proposed approach, not only aiming at reporting state-of-the-art performance. - In Figure 1, the reported perplexities are over 30, which looks pretty high. This high perplexity contradicts better BLEU scores in my experience. How did you calculate perplexity?

Correctness: - In Table 9, the authors uses BT data only from News Crawl 2009, but there should be more recent available monolingual data such as 2010-2014. Why didn't you use these data set? One of the positive effects of BT data is to introduce new training examples with noisy inputs and human-natural outputs, so intuitively the decoder will be updated with diverse topic and data. Why do you think the proposed diversification strategy provides similar improvements as BT data? - Do you think if this approach has regularization effect by augmented data on parallel corpus? - How did you create a vocabulary? is it separate or joint?

Clarity: Tables are located far from the description in the content, e.g. Table 6 appears in page 6 but it was mentioned in page 8. Please fix this allocation as much as possible for readability. Typo: - l. 247 and and learns -> and learns

Relation to Prior Work: None.

Reproducibility: Yes

Additional Feedback: UPDATE: Thank you for providing the authors response. I'm not convinced well with the experimental design and the conclusion that the method complements BT. I decreased my score 6->5.


Review 2

Summary and Contributions: Paper proposes a data augmentation technique for neural machine translation that incorporates the use of multiple models trained on a dataset and harvest the ensembling effect of each of these models, via data diversification. Proposed method starts with training multiple left-to-right and right-to-left models using the supervised data at hand, and then goes into an iterative phase of augmenting more and more data using previous models and at the end training a final left-to-right model. The loop variable "k" determines the diversification factor, by setting how many iterates needs to be completed to augment the final dataset. Paper relates the proposed approach to data augmentation, back-translation, ensemble methods, knowledge distillation and multi-agent dual learning; later provides empirical evidence on multiple datasets, high-mid-low resource MT tasks and at last shares an analysis section laying out the factors playing role in the proposed approach.

Strengths: - Simple approach to implement as an outer loop using any NMT architecture/variant.

Weaknesses: - lack of diversification details (which is hinted in the last section, but not mentioned explicitly) - ignores the translationese effect of forward/backward translation in BLEU scores - although the proposed approach is advertised as a cost efficient solution, the inference cost is not mentioned in the paper, which is expected to be substantially big.

Correctness: 1. Without separating out the translationese effect [1,2] provided BLEU scores will be hard to interpret. Backward-and-forward translation changes (simplifies) the generated output, which is then used to train consecutive models. If at the end, these models are evaluated on test sets that are not natural, but translationese (which is also simpler) then this inflates the BLEU scores. 2. In general the BLEU score improvements are very minor, esp. Table 5. sometimes 0.1 BLEU improvement. In addition, Transformer models are known to have high variance (+-0.5 BLEU) [3], if the proposed approach is not using checkpoint averaging or similar smoothing methods, then the improvements shown in the paper could be spurious. 3. Section 5.1, the formulation ignores the optimization effect, and assumes that the models have perfect fit on the available data [1] https://arxiv.org/abs/1908.05204 [2] https://arxiv.org/abs/1911.03823 [3] https://arxiv.org/abs/1804.09849

Clarity: Paper is clear, easy to read and digest.

Relation to Prior Work: May consider covering the literature on "self-training" [1] which resembles the proposed approach. In terms of data diversification, a similar approach is in [2]. For the model architectures, please consider mentioning hybrid approaches [3] [1] https://openreview.net/pdf?id=SJgdnAVKDH [2] https://arxiv.org/abs/1909.11861 [3] https://arxiv.org/abs/1804.09849

Reproducibility: Yes

Additional Feedback:


Review 3

Summary and Contributions: The paper boosts the performance of NMT by proposing a data diversifying method. It uses forward and backward models to generated samples and merges them with original dataset. The method achieves state-of-the-art BLEU scores in the WMT’14 English-German and English-French translation tasks.

Strengths: The proposed method is a combination of back-translation, ensemble of models, data augmentation and knowledge distillation for NMT. It also gives a detailed analysis of data diversification and conducts a study on hyperparameters and back-translation.

Weaknesses: The novelty of this paper is limited. All the strategies including back-translation are common methods in NMT. This paper only combines them together to achieve high performance without proposing a novel model or architecture. Maybe it’s a very good method to participate in the WMT competition.

Correctness: Yes

Clarity: Yes

Relation to Prior Work: Yes

Reproducibility: Yes

Additional Feedback: Overall, the paper is well written, and model structure and training details are clearly presented. It also gives a detailed analysis of data diversification and conducts a study on hyperparameters and back-translation. However, all the strategies presented in the paper are common methods in NMT, including back-translation and dual learning. In my view, the novelty of this paper has not reached an acceptable level for NIPS conference. There are some missing references: 1. Achieving human parity on automatic chinese to english news translation. Hassan et al. 2018. 2. Iterative Back-Translation for Neural Machine Translation. Hoang et al. 2018. 3. Joint training for neural machine translation models with monolingual data. Zhang et al. 2018. 4. Regularizing neural machine translation by target-bidirectional agreement. Zhang et al. 2019. 5.Synchronous Bidirectional Neural Machine Translation. Zhou et al. 2019. Questions: 1. The proposed method actually can be improved with right-to-left NMT model or bidirectional NMT model, as shown in Hassan et al. (2018). Have you combined the proposed method with right-to-left or bidirectional NMT models? What is the main different between your method and Hassan et al. (2018)?


Review 4

Summary and Contributions: This work describes a simple approach to synthetically augment the training dataset for neural machine translation. The proposed approach involves training multiple forward and backward MT models and appending their outputs on the original training dataset to the training data. This augmented (or diversified) training dataset can then be used to train the next generation of models. 1. The proposed approach is evaluated on the WMT'14 En<->Fr, En<->De, IWSLT En<->Fr, En<->De and Flores En<->Ne and En<->Si tasks, covering a wide range of resource sizes. The approach seems to improve by >1 BLEU over baselines trained directly on the natural dataset on most experiments. 2. The approach is compared against ensembles and baselines trained without data diversification. 3. The approach is shown to improve further when combined with back-translation on additional monolingual data. 4. Additional analysis to demonstrate the effect of the approach on the model's perplexity, ablations to study the effect of varying initial parameters for intermediate models and the effect of including forward translated data on the final model quality. Edit after author response: I would have still liked to see human evaluations to validate the BLEU gains given the extensive use of synthetic data in this paper, but given the additional analysis on translationese I'm updating my score to 6.

Strengths: 1. The described approach is simple, independent of the underlying model architecture and is applicable to any NMT task as long as there's an initial bilingual training set available. The observed improvement on test BLEU is significant. 2. Careful study of various factors, like controlling for parameter initialization, forward translation and perplexity to understand the effect of the approach on model training.

Weaknesses: While the described approach is simple and very generally applicable, there are some major issues with the evaluation that need to be addressed. If 1. and 2. are addressed I would be willing to update my scores. 1. The BLEU evaluation is not clearly described for the WMT and IWSLT experiments. Given the major variations observed in BLEU scores due to differences in post-processing or the BLEU evaluation script used, it's hard to fairly compare against previous work without clearly describing the post-processing, tokenization and BLEU evaluation tool used for these experiments. [1] 2. When training with synthetic data, BLEU scores are an unreliable measure of translation quality due to the translationese effects present in several standard test sets [2,3 and several follow-up works]. Since the proposed method relies heavily on using backward and forward translated data, these effects are bound to affect the observed BLEU improvements. A careful study of the effect of this approach on the forward and backward translated subsets of the evaluation sets and ideally evaluating translation quality with human raters on the two subsets should address this concern. Other minor concerns: 3. Table 1. suggests that the amount of "actual" training data required for Back-translation and NMT+BERT is much higher. This seems misleading given that those approaches rely on unlabeled data which is readily available. 4. Missing reference and discussion on self-training. [4] 5. Additional analysis on out-of-domain generalization (or out-of-domain test sets) would be nice to have. Is augmentation with smooth synthetic data affecting the generalization ability of the model beyond the test sets being used? References: [1] A Call for Clarity in Reporting BLEU Scores, Post et al. [2] APE at Scale and its Implications on MT Evaluation Biases, Freitag et al. [3] On The Evaluation of Machine Translation Systems Trained With Back-Translation, Edunov et al. [4] REVISITING SELF-TRAINING FOR NEURAL SEQUENCE GENERATION, He et al.

Correctness: Yes, no major issues with the method except for the evaluation methodology discussed above.

Clarity: Yes, the paper is easy to read and understand.

Relation to Prior Work: Some missing discussion and references on self-training, but otherwise adequate.

Reproducibility: Yes

Additional Feedback:

[Author Response · NeurIPS 2020]

**All:** We thank the reviewers for their insightful feedback! We feel encouraged that they (R1, R3, R4) appreciate the fact that our method achieves **SOTA** in En-De, En-Fr WMT'14 translation tasks and also outperforms the baselines (*e.g.,* Scaled Transformer) significantly (1-2 BLEU) in 8 other translation tasks in IWSLT and Flores (low-resource) despite being simple and universally applicable (R1, R2, R4) across different NMT architectures. While **all** reviewers appreciated our thorough analysis, some raised concerns regarding the *translationese effect* (R2, R4). In the following, we address this concern along with other specific comments and mischaracterizations.

**1. Translationese effect (R2, R4):** Our method is **NOT** affected by this. To verify, we performed an experiment with the En-De testset provided by Edunov et al., (ACL'20), and compare our method with the Scaled Transformer baseline where both systems were trained on the standard WMT dataset from Vaswani et al., (2017). The table below shows that our method consistently outperforms the baseline in **all 3 scenarios** tested in Edunov et al. Meanwhile, Edunov et al. show that BT outperforms only in $X^* \rightarrow Y$ (source translationese to natural target), thus suffering from the translationese effect, while **our method does not**.

| En-De WMT'14 | $X \rightarrow Y^*$ (natural src $\rightarrow$ translationese tgt) | $X^* \rightarrow Y$ (translationese src $\rightarrow$ natural tgt) | $X^{**} \rightarrow Y^*$ (translationese src $\rightarrow$ translationese tgt) |
|---|---|---|---|
| Baseline | 31.35 | 28.47 | 38.59 |
| Our method | 33.47 | 30.38 | 41.03 |

Our explanation is simple: the BT method of their paper is a **semi-supervised** setup that uses **extra natural monolingual data in the target**. In our method, back-translation is conducted on the translationese part (target side) of the parallel data, and does not enjoy the introduction of natural text which benefits only $X^* \rightarrow Y$. Simply put, back-translation **with** and **without** monolingual data are two different concepts that should not be confused.

In the paper, we sufficiently conducted **9** different analyses (**3** in Appendix) to understand our method better and to distinguish it from BT. It is therefore natural for us to miss the translationese effect. We urge you to allow us to include the translationese analysis and reconsider your decisions.

Reviewer #1 (R1) **(1)** The scaled Transformer is the same as vanilla Transformer, only with more GPUs. The vanilla Transformer is no longer used much for WMT experiments, so we don't think it is necessary to include it, but we can include it if the reviewers disagree with us. **(2)** Perplexity is calculated in the standard way, *i.e.,* the *exponential of cross-entropy*. The perplexity is reversely proportional to BLEU during training (as expected). However, **the best valid set perplexity** (for selecting the model) does not always correlate with the **testset BLEU.** Vaswani et al., (2013) points out that sometimes we get higher BLEU by sacrificing perplexity using methods like temperature-controlled softmax. **(3)** Table 9 is only to show that our method complements BT, not to compete for SOTA (which would require huge compute). The choice of News Crawl 2009 was just random in that aspect. We do not claim our method performs similarly as BT, we claim it complements BT. Introducing **extra monolingual data** gives BT an unfair advantage in comparing with ours. **(4)** Yes, we believe our method has a regularization effect. The vocabulary is joint BPE.

Review #2 (R2) **(1)** We had to put the Diversification details in the Appendix (C) due to page limit. We can bring it back to the main content, if reviewers want. **(2)** There is no extra inference cost compared to standard Transformer. In case you talk about data generation, it costs about 7hrs, which is 30% time to train the baseline. **(3)** About BLEU gains, we compared with SOTA and reported about 1-2 points improvements across the datasets (WMT, IWSLT), which everyone would agree is quite decent if not huge. For your reference, most of the recently published papers (*e.g.,* Shaw et al., 2018; Edunov et al., 2018; Wu et al., 2019) report only 0.1-0.5 BLEU improvements in the standard WMT testset. Table 5 is to show that our approach has an ensemble effect but not to compare with Ensemble as ensembling requires **7x more memory and computations** (thus not fair to compare). And yes, we did **perform checkpoint averaging** (mentioned in Appendix B), so the gains are **not spurious**. **(4)** About Sec. 5.1, we do **not** assume the models have *perfect fit* on the data. In fact, ensembling addresses and regularizes the issue that the models do not have a perfect fit.

Review #3 (R3) **(1) We humbly disagree with this definition of "novelty". Presuming simple methods as non-novel and biasing "novelty" only towards complex architectures can be a hindrance to scientific progress.** NeurIPS should value efforts that are robust, effective and make high impact, yet have not been tried before. Our work is novel as no one has tried using multiple forward & backward models to augment data, and makes high impact as it pushes SOTA by a decent margin. **(2)** We tried bidirectional models but didn't work. Hassan et al. has a semi-supervised setup with one model inducing confidence to the loss of another, while ours are trained independently.

Review #4 (R4) **(1)** The BLEU evaluation is **exactly the same** as previous SOTA papers (Vaswani et al.,2017;Shaw et al., 2018;Edunov et al.,2018; Wu et al.,2019). We use their BPE code and measure tokenized BLEU (Appendix B). Most previous work didn't report SacreBLEU. In case, you want to know SacreBLEU on WMT, here are the numbers with the default detokenized SacreBLEU (*sacremoses* and *sacrebleu* scripts): Scale Transformer gets **28.5** in En-De, **40.9** in En-Fr; our method gets **30.0** in En-De and **41.9** in En-Fr. **(2)** Out-of-domain generalization could be a good extra analysis, where our method has the potential to do better as it is trained on more diversified data. However, notice that we had to leave out many details in the Appendix due to the page limit. We can include it in a later version.

[Meta-Review · NeurIPS 2020]

This work describes a simple approach to synthetically augment the training dataset for neural machine translation. The proposed approach involves training multiple forward and backward MT models and appending their outputs on the original training dataset to the training data. This augmented (or diversified) training dataset can then be used to train the next generation of models. The proposed approach is simple, achieves good results, and the authors do a good job presenting the idea. The paper is quite empirical and the technique fairly specific to NMT, but it is still interesting to see that sometimes simple ideas work well and are thus important / deserve careful consideration. A final request from the AC: it might be better to avoid the word 'elegant' in the title. It's better to let the reader decide whether something is elegant or not and instead stick to more objective adjectives.